# Integrated Transcriptome and Metabolome Analysis Elucidates the Defense Mechanisms of Pumpkin Against Gummy Stem Blight

**DOI:** 10.3390/ijms26062586

**Published:** 2025-03-13

**Authors:** Qian Zhao, Liyan Zhang, Weibo Han, Ziyu Wang, Jianzhong Wu

**Affiliations:** 1Cultivation and Farming Research Institute, Heilongjiang Academy of Agricultural Sciences, Harbin 150086, China; zhaoqian19830401@163.com; 2Forestry College, Inner Mongolia Agricultural University, Huhhot 010011, China; zhangliyanyong@126.com; 3Institute of Forage and Grassland Sciences, Heilongjiang Academy of Agricultural Sciences, Harbin 150086, China; cysbgs2018@163.com (W.H.); wziyu2021@gmail.com (Z.W.)

**Keywords:** *Stagonosporopsis cucurbitacearum*, DAMs, DEGs, transcriptome and metabolome analysis, plant–pathogen interaction

## Abstract

Gummy stem blight (GSB) is a pervasive disease that causes considerable economic losses in cucurbit crops and poses a significant threat to pumpkin production. However, the molecular interaction mechanisms between pumpkin and the pathogen remain largely unexplored. In our previous research, we isolated and identified *Stagonosporopsis cucurbitacearum* (Sc) as the primary causative agent of pumpkin stem blight in Northeast China. Through whole-genome analysis, we identified several pathogenic genes associated with Sc infection in pumpkins. In this study, we performed a comprehensive comparative transcriptomic and metabolomic analysis of unvaccinated and Sc-inoculated pumpkins. We observed distinct differences in gene expression profiles, with these genes being significantly enriched in pathways related to plant–pathogen interactions, phytohormone signal transduction, and metabolic processes, including phenylpropanoid biosynthesis. Joint analysis revealed that the phenylpropanoid biosynthesis pathway was activated in Sc-infected pumpkins. Notably, two metabolites involved in the phenylpropanoid and flavonoid biosynthesis pathways, p-coumaric acid and quercetin, exhibited significant upregulation, suggesting their potential roles in conferring resistance to GSB. These findings enhance our understanding of the molecular mechanisms underlying the defense response against GSB infection in pumpkins and may provide valuable insights for developing strategies to control GSB disease.

## 1. Introduction

Pumpkin is a major cash crop and is grown in almost all regions, from temperate to tropical [1]. According to the statistics from the Food and Agriculture Organisation, global Pumpkin and gourd production reached 22.8 million tons in 2022 (FAOSTAT, www.fao.org/faostat, accessed on 6 December 2023), with a harvest area of 1.52 million hectares. However, the yield and quality of pumpkins are seriously threatened by diseases caused by pathogen attacks.

Gummy stem blight (GSB), caused by *Stagonosporopsis cucurbitacearum* (Sc), is one of the most serious and common pathogens affecting the production of cucurbit plants. GSB can infect the above-ground parts of cucurbit plants, such as leaves, flowers, and fruits, resulting in necrotic spots on the leaves or damage to stems and vines [2,3]. In recent decades, GSB caused by the pathogen *Stagonsporopsis* has become a major disease of field melons and greenhouse cucurbits, with yield losses reaching up to 30% at the seasonal peak, which significantly impacts global cucurbit crop production [4,5,6,7]. Currently, chemical control, particularly fungicides, is the most widely used method to control GSB. However, excessive application of fungicides inevitably causes negative impacts on the environment and food safety. Additionally, their effectiveness is declining due to the increasing resistance of certain pathogenic isolates to these chemicals [8,9].

In the long-term struggle between plants and pathogenic fungi, plants have evolved elaborate and complex immune systems. These systems activate plant resistance by recognizing pathogen effectors, thereby protecting themselves from attack [10]. As pathogens continue to evolve and plant defense systems degrade, it is crucial to study how plants recognize and defend against these attacks through their complex immune systems [11,12]. Clarifying the host’s defense response to pathogen infection is essential for understanding the mechanisms of disease resistance. High-throughput omics technologies have become powerful tools for studying plant defense responses to biotic stresses. Among these, transcriptomics is widely used to identify genes, signal-transduction pathways, and regulatory networks involved in plant–pathogen interactions. In previous studies, our team used transcriptomics to demonstrate that an apyrase-like gene plays a significant role in the defense response of pumpkin to GSB [13]. With the advancement of science and technology, a single omics approach has become insufficient for meeting the needs of scientific research. Synomics, as a hotspot for promoting scientific research, has garnered widespread attention, and disease research has also shown a trend towards multi-omics development. The interactions among genes, mRNAs, and regulatory factors in organisms form a network relationship. To clarify the regulation and causality between each molecule, it is necessary to construct a gene regulatory network to link them together. This will provide a deeper understanding of the molecular mechanisms and genetic basis of complex traits in genetic diseases.

The combined analysis of transcriptome and metabolome offers the possibility of characterizing plant–pathogen interactions and pathogenesis at the molecular level [14]. Fu Rongtao et al. combined transcriptomic and metabolomics analyses to reveal the pathogenic factors of *Ustilaginoidea virens*. They found some differentially expressed genes (DEGs) involved in mitophagy, secondary metabolism, and amino acid metabolism, along with differentially accumulated metabolites (DAMs) such as alanine, tyrosine, histidine, methionine, cysteine, and fatty acids, which are closely linked to the pathogenicity of *U. virens* [15]. Wei Ye et al. indicated that pathogenicity-related genes, especially the gene encoding the loss-of-pathogenicity B (LopB) protein, cell wall-degrading enzymes (particularly glycosyl hydrolase-related genes), and killer and Ptr necrosis toxin-producing related unigenes in *Bipolaris sorokiniana* from wheat leaf, play significant roles in its pathogenicity [16]. The exploration and analysis of disease-resistance-related genes and metabolites will lay the core theoretical foundation for the design of durable new disease-resistant varieties and the development of green agriculture.

Breeding resistant cultivars is one of the most efficient approaches for disease control. Several studies have explored germplasm resources resistant to blight [17,18]. For instance, Zhang Y. et al. conducted screenings of melon (*Cucumis melo*) for resistance to gummy stem blight in both greenhouse and field settings. They evaluated 798 U.S. Department of Agriculture Plant Introduction (PI) accessions and 24 related *Cucumis* species, identifying two highly resistant varieties and several moderately resistant ones [17]. Similarly, Wolukau, J.N. et al. analyzed 200 germplasm sources screened against a highly virulent isolate of *Didymella bryoniae* in a plastic tunnel. They identified a wild *Cucumis* from the highlands of Kenya as highly resistant, along with multiple other resistant varieties [18]. Applying resistance factors identified through multi-omics approaches to the identification, utilization, and creation of resistant germplasm can provide crucial biological information for developing new disease and pest control methods and reducing the application of chemical pesticides. This aligns with broader efforts in modern breeding programs to enhance crop resistance and sustainability.

Before the emergence of metabolomics, the development of genomics, transcriptomics, and proteomics significantly advanced our understanding of plant diseases and the mechanisms that determine whether a pathogen can successfully obtain nutrients and evade plant immunity. Genomic studies, which analyze the genetic architecture of both plants and pathogens, have been instrumental in monitoring how organisms adapt to disease pressure [19,20]. Transcriptomic studies have provided insights into which host genes are manipulated by pathogens in a disease setting or are reprogrammed to facilitate a successful defense response.

The occurrence of GSB is becoming increasingly severe, posing a significant threat to *cucurbitaceae* crops such as pumpkin. Currently, progress in exploring resistance resources, cloning resistance genes, and identifying key pathogenic factors in squash resistant to GSB is limited. Studies on the gene function of anti-bacterial wilt can elucidate a series of molecular mechanisms at the molecular level, including the infection and pathogenic processes, as well as the pathogen–host interaction mechanisms. These insights play an important role in understanding the occurrence and prevention of anti-bacterial wilt. In this study, based on the identification of genes related to GSB resistance in pumpkin and the acquisition of data information, we mined genes associated with GSB resistance and analyzed the resistance mechanism. This provided crucial biological information for breeding GSB-resistant specific varieties.

## 2. Results

### 2.1. Transcriptome Analyses

#### 2.1.1. Sequencing Data and Quality Analysis

The transcriptome analysis included a total of six samples, with three samples each from the ZQ and CK groups. Utilizing the Illumina HiSeq™2500 second-generation high-throughput sequencing platform and the PE150 sequencing strategy, a total of 36.39 million raw reads were obtained. After quality control, the clean reads averaged 35.95 bp in length, with Q20 ranging from 97.54% to 97.87%, Q30 from 93.26% to 94.03%, and GC content from 46.36% to 47.40% (Appendix A). The STAR (Spliced Transcripts Alignment to a Reference) software 2.7.9 was employed to align the reads to the reference genome (reference genome: https://www.ncbi.nlm.nih.gov/assembly/GCF_002738345.1/, accessed on 11 February 2025). Over 95.68% of the sequencing data from the samples were successfully aligned to the reference genome (Appendix A). The sequencing data collected in this study are suitable for further bioinformatics analysis. RNA-seq data of this study can be found at the National Center for Biotechnology Information (https://www.ncbi.nlm.nih.gov/bioproject/) with the BioProject ID PRJCA036965.

#### 2.1.2. Identification and Functional Enrichment Analysis of DEGs

Based on the Pearson correlation coefficient R [21], the correlation coefficients between ZQ1 and ZQ2, ZQ1 and ZQ3, and ZQ2 and ZQ3 were 0.9734, 0.98, and 0.9791, respectively (Appendix A). In contrast, for the three replicate libraries of CK, the coefficients between CK1 and CK2, CK1 and CK3, and CK2 and CK3 were 0.9649, 0.7138, and 0.759, respectively (Appendix A). The Pearson correlation coefficient quantifies the strength of a linear relationship, with values between 0.8 and 1.0 indicating a strong correlation, and values between 0.6 and 0.8 indicating a moderate correlation. Consequently, the three replicate libraries of both ZQ and CK were chosen for further analysis. Differential expression analysis between the groups identified 1176 DEGs within the ZQ replicates, comprising 610 downregulated and 566 upregulated genes (Figure 1A, Table 1). Hierarchical clustering of these DEGs based on log10 (RPKM) values revealed that the overall gene expression patterns could be categorized into distinct clusters according to the expression levels of the DEGs across the six samples (Figure 1B).

GO enrichment analysis revealed that the DEGs were associated with 55 GO terms at level 1. Notably, four biological process terms were specifically enriched post-infection, which included the cellular process, metabolic process, single-organism process, and response to stimulus (Figure 2A). At level 2 of the GO enrichment, four specific biological process terms were identified as enriched: cell wall organization (GO:0010383), wall polysaccharide metabolic process (GO:0044264), cellular polysaccharide metabolic process (GO:0010411), and xyloglucan metabolic process (GO:1901700) (Figure 2B). Further investigation highlighted several genes annotated as xyloglucan endotransglucosylase/hydrolase (XTH), such as *ncbi11146684*, *ncbi111485079*, *ncbi111469155*, *ncbi111491143*, *ncbi111492413*, and *ncbi111468264*. Among these, the gene ncbi111485079 was present in all four GO categories and may be implicated in disease resistance.

To further understand the functional responses of DEGs to cranberry blight, a KEGG enrichment analysis was conducted. This analysis identified 98 pathways, revealing that the genes differentially expressed in pumpkin (ZQ) were significantly enriched in several pathways following infection. These pathways are associated with the biosynthesis of secondary metabolites, protein processing in the endoplasmic reticulum, cutin, suberin, wax biosynthesis, plant hormone signal transduction, photosynthesis, starch and sucrose metabolism, phenylpropanoid biosynthesis, metabolic pathways, fatty acid elongation, and the MAPK-signaling pathway in plants (Figure 3A). These findings suggest that pumpkin (ZQ) mounts a distinct defense response against GSB by modulating various pathways.

#### 2.1.3. Novel Gene Mining and Functional Annotation

A total of 858 novel genes were discovered, with 35 of these being differentially expressed (Appendix A). GO and KEGG enrichment analyses were conducted on these new DEGs. The GO processes of particular interest included oxidation–reduction process (GO:0055114), response to auxin (GO:0009733), organic acid metabolic process (GO:0006082), galactose metabolic process (GO:0006012), carbohydrate metabolic process (GO:0005975), and again, response to auxin (GO:0009733). Additionally, the GO functions that garnered more attention included molecular function (GO:0003674), catalytic activity (GO:0003824), endoribonuclease activity (GO:0004521), and xylulokinase activity (GO:0004856), among others. Furthermore, five genes were annotated to specific KEGG pathways, such as the MAPK-signaling pathway plant (MSTRG.7283), plant hormone signal transduction (MSTRG.8150), biosynthesis of secondary metabolites (MSTRG.8760), folding, sorting, and degradation (MSTRG.16392), and Brassinosteroid biosynthesis (MSTRG.21397).

#### 2.1.4. Transcription Factor Analysis

In this study, a total of 2320 transcription factors were identified, encompassing 56 transcription factor families, such as bHLH, ERF, MYB, C2H2, NAC, WRKY, bZIP, and others (Figure 4A). GO and KEGG enrichment analyses highlighted changes in the expression levels of 116 transcription factor genes, which were distributed across 23 transcription factor families, predominantly MYB, bZIP, ERF, bHLH, NAC, WRKY, and so on (Figure 4B, Appendix A). Among these, 54 transcription factors were upregulated and 62 were downregulated. Notably, five genes from the WRKY family were significantly enriched, specifically *ncbi111480688*, *ncbi111481142*, *ncbi111488796*, *ncbi111492721*, and *ncbi111495725*. Additionally, seven genes from the NAC family were significantly enriched, including *ncbi111465021*, *ncbi111477085*, *ncbi111483064*, and others. There were 16 genes from the AP2/ERF sub-family that showed significant enrichment, comprising ERF (*ncbi111468308*, *ncbi111469073*, *ncbi111470378*, and 15 other genes) and RAV (*ncbi111472203*). In our study, 13 bHLH DEGs (*ncbi111466828*, *ncbi111469692*, *ncbi111474286*, etc.) and 15 bZIP DEGs (*ncbi111468251*, *ncbi111470444*, *ncbi111473996*, etc.) were also enriched.

In conjunction with KEGG pathway analysis, it was found that the pathways for plant–pathogen interaction (*ncbi111484520*) and signal transduction (*ncbi111470444*, *ncbi111482940*, *ncbi111484520*, *ncbi111485164*, *ncbi111486047*, *ncbi111487159*, *ncbi111489632*, and *ncbi111499095*) were significantly enriched. It is speculated that these genes play a crucial role in the pumpkin’s resistance to GSB.

#### 2.1.5. qRT-PCR Validation

To confirm the reliability of the RNA-seq digital expression levels, 10 DEGs were selected for validation using qRT-PCR. These genes included *ncbi111472985*, *ncbi111478197*, *ncbi111486983*, *ncbi111493168*, *ncbi111482187*, *ncbi111470962*, *ncbi111499896*, *ncbi111480526*, *ncbi111484320*, and *ncbi111485296*. A linear correlation analysis was conducted between the qRT-PCR results and the transcriptome data. The analysis revealed a correlation coefficient R^2^ of 0.9268, indicating a strong correlation between the two datasets (Appendix A). The expression patterns of these genes, as determined by qRT-PCR, closely matched those observed in the transcriptome data, thereby validating the accuracy of the RNA-seq results (Figure 5).

### 2.2. Metabolome Analyses

#### 2.2.1. Reliability Analysis of Metabolomics Data

The transcriptome analysis involved six samples from the ZQ group and six samples from the CK group. Quantitative analysis of the correlation data between samples revealed a high degree of similarity in metabolic composition and abundance, confirming that the samples were suitable for subsequent analysis (Appendix A). Principal component analysis (PCA) was applied to the ZQ and CK samples to discern overall differences and the extent of changes in metabolites between the two groups. The results showed minimal quality control (QC) differences, as indicated by the tight clustering of QC samples on the PCA plot, which suggests high methodological stability and reliable data quality. A distinct separation trend between the groups was observed, with significant differences noted between the ZQ and CK groups (Appendix A). The separation between groups was minimal, and the sample reproducibility was excellent. The dense distribution of QC samples in the PCA analysis diagram further confirms the reliability of the data.

To maximize the intergroup differences between the healthy and susceptible groups, subsequent model testing and differential metabolite screening were conducted using OPLS-DA results. The classification performance of the model was assessed using the R^2^X, R^2^Y, and Q^2^ predictive parameters, as well as the OPLS-DA score plots. As shown in the figures, the scores in negative ion mode are: R^2^X = 0.623, R^2^Y = 0.994, Q^2^ = 0.952, and RMSEE = 0.046 (Figure 6A). In positive ion mode, the scores are: R^2^X = 0.46, R^2^Y = 0.993, Q^2^ = 0.944, and RMSEE = 0.048 (Figure 6B). These results indicate that the R^2^Y and Q^2^ values are above 0.9, which signifies an excellent model fit, while the RMSEE value remains below 0.048. This demonstrates the high reliability of the model, thereby validating its suitability for further analysis.

#### 2.2.2. Identification and Functional Enrichment Analysis of DAMs

To elucidate the metabolic alterations that occur after Sc infection, untargeted metabolomics was conducted to assess the metabolite profiles in healthy and diseased leaves at 72 h post-infection. In this analysis, 67 differential metabolites were identified using the NEG method, comprising 14 upregulated and 53 downregulated metabolites. Additionally, the POS method detected a total of 133 differential metabolites, with 34 being upregulated and 99 downregulated (Appendix A). By employing the Z-score (standard score) to quantify the relative content of metabolites at the same level, it was found that Trehalose, Trans-3′-hydroxycotinine o-beta-d-glucuronide, Goitrin, Pro-ser, and Adenosine were significant DAMs (Figure 6C,D).

In contrast to the whole background, KEGG enrichment analysis identified 51 DAMs involved in the defense response of pumpkin to GSB using the NEG method, which could be categorized into 12 classes (Appendix A). Furthermore, 48 DAMs were identified using the POS method and were also divided into 12 classes (Appendix A). The results indicated that amino acid metabolism, the biosynthesis of other secondary metabolites, carbohydrate metabolism, and energy metabolism were specifically enriched. Additionally, nine pathways were specifically enriched after infection in the NEG method, including carbohydrate metabolism (galactose metabolism), membrane transport (ABC transporters), the biosynthesis of secondary metabolites (biosynthesis of unsaturated fatty acids and fatty acid biosynthesis), peptide metabolism (glutathione metabolism), lipid metabolism (fatty acid elongation, fatty acid degradation, and fatty acid metabolism), and amino acid metabolism (taurine and hypotaurine metabolism) (Figure 3B). On the other hand, only one pathway was specifically enriched after infection in the POS method, which is ABC transporters (Figure 3C). These findings demonstrate that pumpkin exhibits contrasting defense responses to GSB by modulating various metabolic pathways.

### 2.3. Integration of Transcriptome and Metabolome Profiles

To gain a comprehensive understanding of the pumpkin’s defense response to GSB at both the transcriptomic and metabolic levels, an integrated analysis of transcriptome and metabolome was conducted. A total of 43 KEGG pathways, belonging to 11 KEGG categories, were found to be enriched in transcriptome and metabolome after infection, such as amino acid metabolism, the biosynthesis of other secondary metabolites, carbohydrate metabolism, energy metabolism, global and overview maps, lipid metabolism, membrane transport, metabolism of cofactors and vitamins, metabolism of other amino acids, metabolism of terpenoids and polyketides, and nucleotide metabolism. Among these, 24 KEGG pathways were significantly enriched, and there were notable differences in their gene expression profiles. The altered genes were particularly enriched in terms related to the biosynthesis of secondary metabolites, cutin, suberin and wax biosynthesis, starch and sucrose metabolism, phenylpropanoid biosynthesis, and more (Appendix A).

To ascertain the relationship between DEGs and DAMs in host squash following paraquat infection, and to address the issues of data loss and noise that arise when analyzing single omics data, an integrated analysis of transcriptome and metabolome data was conducted. This approach minimizes the occurrence of false positives, facilitating the study of phenotypic changes and the regulatory mechanisms of biological processes in biological models. To identify the interrelations between metabolites and genes, we generated different omics loadings plots for variables in the overlapping part (Figure 7A). Based on the loading values of the elements, we identified the top 250 genes and metabolites with the highest squared loading values in the first two dimensions of the integrated loading map, thereby highlighting the genes and metabolites with the strongest correlations (Appendix A). The metabolites included M601T34 POS (17-epimethandienone), M152T254 POS (Guanine), M401T358 NEG (Gentiopicroside), and M855T187 POS (Ophiopogonin), while the top five genes included *ncbi111482302*, *ncbi111472194*, *ncbi111493021*, *ncbi111499505*, and *ncbi111484985*, which may play significant roles in resistance to pathogen infection (Appendix A). Additionally, the Pearson correlation coefficient was employed to assess the correlation between differential genes and differential metabolites. According to the Pearson coefficient, the top 20 differential metabolites closely associated with differential genes were identified, including M173T123 NEG (Dehydroascorbic acid), Octopine, M303T417 POS (Morin), M391T234 POS (Mitraphyllin), and M163T391 POS (L-(-)Sorbose) (Appendix A).

To identify DEGs associated with phytoalexin synthesis in pumpkin leaves infected with Sc, an integrated analysis of the phenylpropanoid and flavonoid biosynthesis pathways was conducted using transcriptome and metabolome data. The DEGs and DAMs from our omics datasets were mapped to the KEGG pathways and visualized as a network of genes and metabolites (Figure 7B). Several genes encoding enzymes involved in phenylpropanoid and flavonoid biosynthesis were significantly regulated following Sc infection. In this study, genes such as phenylalanine ammonia-lyase (PAL), shikimate O-hydroxycinnamoyl transferase (HCT), cinnamoyl-CoA reductase (CCR), beta-glucosidase (bglX), and peroxiredoxin 6 (PRDX6) exhibited differential expression after Sc infection (Figure 7B). Specifically, the beta-glucosidase (bglX) gene was upregulated, while peroxiredoxin 6 (PRDX6) and phenylalanine ammonia-lyase (PAL) showed both upregulation and downregulation patterns; the other genes were downregulated. Additionally, metabolites such as Phenylalanine, L-Tyrosine, p-Coumaric acid, Cinnamic acid, Quercetin, Caffeoyl quinic acid, Scopoletin, and Syringin were differentially expressed post-infection, with p-Coumaric acid and Quercetin showing particularly significant changes in expression.

## 3. Discussion

In recent years, the production of pumpkins in China has become increasingly industrialized, but continuous cropping has led to a rise in the severity of GSB disease, causing significant losses in pumpkin production [21,22]. This pathogen has been documented across six continents, affecting at least 12 genera and 23 species of Cucurbitaceae [4]. While research on GSB resistance mechanisms has achieved breakthroughs in crops such as watermelon (*Citrullus lanatus*) [23], cucumber (*Cucumis sativus* L.) [24], pumpkins (*Cucurbita* spp.) [25], muskmelon (*C. melo* L.) [26], and gourds (*Lagenaria siceraria* (Molina) Standl) [27], the specific mechanisms of GSB resistance in pumpkins remain largely unknown. Additionally, the pathogen’s host is widely distributed, and its structure is complex and varied, making GSB control challenging [28]. To date, there have been few studies on the epidemiology of GSB in *Cucurbitaceous* plants [29,30,31,32], which could aid in developing effective prevention and control methods to reduce the incidence of GSB in pumpkins. In this study, we investigated the defense responses of pumpkins to GSB using transcriptome and non-targeted metabolome analyses. We characterized DEGs, transcription factors, and DAMs that are involved in the pumpkin’s defense response to GSB.

### 3.1. Transcriptomic Analysis

Transcriptomics has provided insights into which host genes are manipulated by pathogens in a disease setting or are reprogrammed for a successful defense response. From a transcriptomic perspective, pumpkins exhibit differences in gene expression in response to Sc infection. The Sc strain possesses numerous genes associated with fungal/pathogen defense mechanisms and shows a higher concentration of DEGs during infection (Figure 1). Plants activate defense mechanisms to protect themselves when attacked by pathogens, responding swiftly with either direct or indirect reactions. Xyloglucan endotransglucosylase/hydrolase (XTH) enzymes are not strictly necessary for cell wall loosening during plant cell expansion but play crucial roles in response to specific biotic or abiotic stresses [33,34]. In this study, we observed that some XTH genes were enriched and identified within specific biological process terms. These genes may be associated with cell wall biogenesis and organization. Notably, the gene *ncbi111485079*, which may serve as a dynamic barrier against pathogen invasion, is potentially linked to disease resistance (Figure 2B).

### 3.2. Transcription Factors Analysis

Transcription factors (TFs) modulate gene expression by binding to the promoter regions of target genes, playing an essential role in plant growth, stress tolerance, and the pathogenicity of plant pathogens [34]. Enrichment analysis of GO and KEGG has identified various families of TFs, including MYB, bZIP, ERF, bHLH, NAC, WRKY, and HD-ZIP, among others (Figure 4).

WRKY transcription factors are involved in both PAMP-triggered immunity (PTI) and effector-triggered immunity (ETI) at multiple regulatory levels [35]. They can interact with PAMPs or effector proteins to either activate or inhibit PTI-related biological processes. The responsiveness of several WRKY genes to pathogens, elicitors, and defense-related phytohormones indicates a significant role for this family in plant immunity [36]. In this study, five WRKY family genes were identified that may be involved in the biosynthesis of defense compounds and defense responses, suggesting potential links between these transcription factors and the biological processes underlying disease resistance.

NAC genes have been identified through genome-wide studies and expression analyses in various plant species such as rice, tomato, tobacco, and cucumber [37,38,39,40]. NAC transcription factors related to immunity, belonging to different NAC subfamilies, have been reported to play crucial roles in plant immunity as negative or positive regulators, modulators of the hypersensitive response (HR), and stomatal immunity, or as targets of pathogen effectors [41]. In this study, seven DEGs were noted as NAC transcription factors, indicating that NAC genes may be involved in defense responses against GSB invasion.

AP2/ERF transcription factors have been shown to play vital roles in plant developmental processes, tolerance to both biotic and abiotic stresses, and hormone signal transduction [42,43]. ERF genes, acting as terminal response genes in the ethylene-signaling pathway, are involved in regulating the biosynthesis of various plant hormones, including ethylene, auxin, cytokinin, gibberellin, ABA, and jasmonate. In this study, 15 ERF genes were found to be enriched, suggesting their potential contribution to stress tolerance and hormone signal transduction in plants following GSB infection.

bHLH proteins have been extensively studied in various plants, including capsicum annum, tobacco, potato, and tomato [44,45,46,47]. Research indicates that bHLH transcription factors play a significant role in numerous physiological processes such as plant growth and development, metabolism of secondary substances, and responses to abiotic stress. bHLH factors regulate the expression of related genes through the mutual recognition and interaction between their specific functional domains and target genes, thereby influencing plants via metabolic-signaling pathways. In this study, 13 bHLH transcription factors were identified that may be associated with immune responses against GSB. These factors may have a central role in many metabolic processes and are linked to both primary and specialized plant metabolites.

Genetic and molecular studies have demonstrated that bZIP transcription factors in plants regulate a wide array of biological processes, including responses to both abiotic and biotic stresses [48]. The role of bZIP transcription factors in plant defense has been established in species such as arabidopsis, where two out of ten bZIP groups have been shown to contribute to plant innate immunity [49], as well as in tobacco and maize, where certain members of these groups are thought to be involved in defense responses [50,51]. TGA family bZIP transcription factors modulate the expression of PR genes through their physical interaction with the positive regulator nonexpresser of PR genes 1 (NPR1), as reported by Kesarwani et al. [51]. In our study, the genes *ncbi111482940* and *ncbi111489632* were annotated as TGA proteins. Among the bZIP transcription factors, the extensively studied TGA proteins are central to signaling pathways mediated by salicylic acid (SA) and to defense mechanisms against pathogen attacks [52].

Transcription factors (TFs), through their sequence-specific interactions with cis-regulatory DNA elements in gene promoters, emerge as key regulators of plant defense responses against a wide array of pathogens associated with significant diseases. They are part of a complex network involving cross-talk between different signal transduction pathways. Consequently, genes encoding TFs act as master regulators of stress-related genes and present extensive opportunities for engineering pathogen resistance in plants. They are promising candidates for pumpkin breeding, leveraging the molecular techniques that have recently been emerging and applied to plant breeding in the era of genomics and genome editing.

### 3.3. Metabolome Analysis

Metabolomics offers not only a qualitative and quantitative method for determining the pathogenesis of pathogenic fungi but also aids in elucidating the defense mechanisms of their host plants, serving as a powerful tool for decoding plant–pathogen interactions [53]. This approach can detect a range of metabolites associated with infection, such as molecules secreted by pathogens during colonization [54], or amino acids and sugars whose production is induced or mislocalized in the host to facilitate pathogen growth. In our study, we found significant accumulation in several metabolic pathways, including amino acid metabolism, the biosynthesis of other secondary metabolites, carbohydrate metabolism, global and overview maps, membrane transport, and metabolism of cofactors and vitamins.

Metabolites play a variety of roles in plant–pathogen interactions, including surveillance against pathogen attack, signal transduction, enzyme regulation, cell-to-cell signaling, and antimicrobial activity [55]. Disease infection can cause significant perturbations in plant metabolism. It is well established that plant secondary metabolites, such as phenolic compounds, alkaloids, glycosides, and terpenoids, play crucial roles in plant–pathogen interactions. In this study, nine pathways were specifically enriched after infection, including carbohydrate metabolism (galactose metabolism), membrane transport (ABC transporters), biosynthesis of secondary metabolites (biosynthesis of unsaturated fatty acids and fatty acid biosynthesis), peptide metabolism (glutathione metabolism), lipid metabolism (fatty acid elongation, fatty acid degradation, and fatty acid metabolism), and amino acid metabolism (taurine and hypotaurine metabolism) (Figure 3B,C). Compared to the whole background, KEGG enrichment analysis revealed that amino acid metabolism, the biosynthesis of other secondary metabolites, carbohydrate metabolism, and energy metabolism were specifically enriched. These results are expected to provide insights into the defense response of pumpkins to GSB at the metabolic level and offer valuable information for elucidating the resistance mechanisms of plant hosts to GSB.

Plant metabolism is crucial in determining the outcomes of attempted infections. Metabolomics has the potential to uncover disturbances in signaling or output pathways that play key roles in shaping the outcomes of plant–microbe interactions. However, the application of this omics approach and its associated tools in plant pathology research lags behind genomic and transcriptomic methods. Therefore, it is essential to harness the power of metabolomics to advance the study of plant resistance [56].

### 3.4. Metabolome and Transcriptome Analysis

The integration of metabolomics data with transcriptome data enriches the study of plant–pathogen interactions by providing additional layers of information. This includes the identification of metabolites with antimicrobial properties [54], differences in metabolomic profiles between infected and non-infected plants [57,58,59], and the infection and colonization of pathogens [58,60].

Combined transcriptome and metabolome analysis revealed a differential accumulation of metabolites associated with phytohormone signaling, phenylpropanoid metabolism, flavonoid biosynthesis, and nicotinate and nicotinamide metabolism, highlighting the involvement of these pathways in defense responses against pathogenic infections [61,62]. In this study, the correlation analysis between the metabolome and transcriptome identified ABC transporters, alanine, aspartate and glutamate metabolism, alpha-linolenic acid metabolism, and amino sugar and nucleotide sugar metabolism as potential defense responses to pathogenic infections. Additionally, 26 DAMs were found to be connected to DEGs after GSB infection in pumpkins. Notably, M145T298NEG (p-Coumaric acid) and M303T300POS (Quercetin) were observed, which are involved in phenylalanine metabolism and flavonoid biosynthesis, respectively. This suggests that these two metabolites and their related genes may be closely associated with defense responses against GSB infection (Figure 7B).

Additionally, we observed that L-(-)Sorbose, an intriguing natural sugar, plays a significant role in plant stress responses. It has been established that enriching carbohydrate metabolism with a sorbitol branch enhances plant fitness under stress. Sorbitol biosynthesis acts as an electron sink by regenerating NADPH/H+ to NADP+, which, during stress, helps alleviate high electron pressure from photosystems and, thus, reduces the risk of excessive reactive oxygen species (ROS) formation [63]. Concurrently, the gene *ncbi111480972* exhibited catalytic activity in GO analysis and was associated with amino acid metabolism in KEGG, showing a strong correlation with L-(-)Sorbose. This suggests that *ncbi111480972* may play a crucial role in the host’s resistance to infection.

Phenylalanine ammonia-lyase (PAL) is a well-studied enzyme in plant biology due to its pivotal role in linking primary metabolism to secondary phenylpropanoid metabolism, and it plays significant roles in plant–pathogen interactions [64,65]. The phenylpropanoid pathway commences with three key enzymes: phenylalanine ammonia-lyase (PAL), hydroxycinnamoyl transferase (HCT), and 4-coumarate-CoA ligase (4CL). In this study, we found that phenylalanine ammonia-lyase (PAL) was differentially expressed following Sc infection (Figure 7B). Muskan Amjad et al. have suggested a potential role for PAL genes in the plant’s response to various stress conditions and tolerance to both biotic and abiotic stresses [66,67]. Shikimate hydroxycinnamoyl transferase (HCT) catalyzes the transfer of the p-coumaroyl group from p-coumaroyl-CoA to shikimate [68]. Two HCT genes exhibited contrasting expression patterns before and after infection, with *ncbi111478230* and *ncbi111488196* showing decreased expression post-infection. Other genes in these related pathways, such as cinnamoyl-CoA reductase (CCR) (*ncbi111488801*), displayed consistent trends and were upregulated after infection (Figure 7B).

### 3.5. Functional Validation of Novel Genes in Plant Defense Mechanisms

To validate the functional roles of novel genes in plant defense, techniques such as CRISPR-Cas9 gene editing or RNA interference (RNAi) are widely used [69]. For instance, knockout mutations in defense-related genes have been demonstrated to change the susceptibility to pathogens in crops such as *Arabidopsis* [70]. Recent studies have also utilized CRISPR-Cas9 to validate genes involved in resistance to fungal pathogens, including *Didymella bryoniae*, the causal agent of gummy stem blight in watermelon. For example, the silencing of a necrosis-related gene (NRL) in watermelon significantly reduced susceptibility to gummy stem blight, highlighting the gene’s role in basal resistance [71].

Comparative genomics approaches have been successfully applied to identify conserved and divergent defense mechanisms across plant species. A comparison of the genomes of *Cucumis sativus* and *Cucumis melo* has revealed that many defense-related gene families are conserved within the cucurbitaceae family, while others exhibit lineage-specific expansions [72]. OrthoMCL software v5.0 has been widely used for ortholog identification and gene family analysis [73]. Such comparative analyses could provide insights into the evolutionary pressures shaping defense systems in the studied species. Comparing the genomes of watermelon (*Citrullus lanatus*) and melon (*Cucumis melo*) identified conserved signaling pathways associated with gummy stem blight resistance, including the NPR1-mediated salicylic acid pathway [74].

While this study identifies promising candidate genes for plant defense, several limitations persist. For instance, experimental designs featuring larger sample sizes and longer time-series data could more effectively capture the dynamics of gene expression under natural infection scenarios [75]. Functional validation of these novel genes across diverse genetic backgrounds and environmental conditions is also essential to confirm their roles in defense. Recent advancements in metabolomic profiling have revealed that secondary metabolites, such as *cucurbitacins*, play a critical role in resistance to gummy stem blight [76]. Future research could focus on characterizing the regulation of these compounds and their interactions with fungal pathogens.

The identification of novel defense-related genes holds significant importance for agriculture. For instance, marker-assisted breeding or genetic engineering could utilize these genes to develop crops with enhanced pathogen resistance [77]. Mapping these genes to quantitative trait loci (QTLs) associated with disease resistance could further enhance their practical applications. Future research could also explore how these genes are regulated at the transcriptional and post-transcriptional levels, potentially identifying targets for precision agriculture. In the case of gummy stem blight, recent progress in genome editing has enabled the development of watermelon and melon varieties with enhanced resistance to *Didymella bryoniae*. Engineered watermelon plants with an overexpression of a fungal resistance gene had significantly reduced disease severity under field conditions [78]. These advancements highlight the potential for genetic engineering to address this major threat to cucurbit production.

## 4. Materials and Methods

### 4.1. Plant Materials and Artificial Inoculation

In this study, the plant material used was JinBei No.1, a variety of *Cucurbita pepo* from Heilongjiang Province, China. The seeds were first sterilized with 70% ethanol and then sown in pots filled with a sterilized peat–perlite substrate at a ratio of 2:1 (*v*/*v*). These pots were placed in a greenhouse, where seedling management followed standard commercial practices. At the third true leaf stage, uniform and healthy seedlings were selected for subsequent experiments.

Pathogenic fungi were isolated from pumpkin stems exhibiting typical GSB symptoms and were identified as Sc [31,79]. The purified fungi were inoculated onto potato dextrose agar culture medium and incubated at 25 °C in the dark for 3 days. The mycelium was then inoculated onto the leaves of the pumpkin. After inoculation, the seedlings were covered with plastic wrap to maintain a relative humidity above 75%, with temperatures ranging from 25 °C to 30 °C. Simultaneously, sterilized distilled water was sprayed on the other seedlings to serve as controls. A completely randomized block experimental design was employed, with three biological replicates for both treatments and controls. Each replicate consisted of 20 seedlings, and pathogens were separated from pumpkin leaves using cellophane for transcriptome and metabolome analyses. Leaves were sampled 5 days post-inoculation, immediately frozen in liquid nitrogen, and stored at −80 °C for subsequent analyses.

### 4.2. Transcriptome Analysis

#### 4.2.1. Sampling and Experimental Design

The samples were immediately frozen and stored in liquid nitrogen until analysis. Total RNA was isolated using the Trizol Reagent (Invitrogen, Carlsbad, CA, USA. Catalog No. 15596-026) followed by purification with the RNeasy MinElute Cleanup Kit (Qiagen, Hilden, North Rhine-Westphalia, Germany. Catalog No. 74204). RNA degradation and contamination were monitored using 1% agarose gels. RNA purity was assessed using the NanoPhotometer^®^ spectrophotometer (Implen, Munich, Germany). The RNA was purified, and its concentration was measured using the Qubit^®^ RNA Assay Kit in the Qubit^®^ 2.0 Flurometer (Life Technologies, Carlsbad, CA, USA). RNA integrity was evaluated using the RNA Nano 6000 Assay Kit on the Agilent Bioanalyzer 2100 system (Agilent Technologies, Santa Clara, CA, USA).

#### 4.2.2. Library Construction and RNA-seq and Quality Control

For each replication, 3 μg of RNA were used to prepare the sequencing libraries. The libraries were generated using the NEBNext^®^Ultra™ RNA Library Prep Kit for Illumina^®^ (NEB, Ipswich, MA, USA) following the manufacturer’s recommendations. Index codes were added to attribute sequences to each sample. The clustering of the index-coded samples was performed on a cBot Cluster Generation System using the TruSeq PE Cluster Kit v3-cBot-HS (Illumina, San Diego, CA, USA) according to the manufacturer’s instructions. After cluster generation, the library preparations were sequenced on an Illumina Hiseq 2000 platform, and paired-end reads were generated. The raw data (raw reads) in fastq format were initially processed using in-house Perl scripts. Concurrently, the Q20, Q30, GC-content, and sequence duplication levels of the clean data were calculated.

#### 4.2.3. Differential Expression, Gene Annotation and Novel Gene

Differential expression analysis was conducted using the DESeq R package (version 1.10.1). Genes with an adjusted *p*-value of less than 0.05 were identified as differentially expressed. Gene ontology (GO) enrichment analysis of these DEGs was performed using the topGO R package 2.4.5, based on the Kolmogorov–Smirnov test [80]. Additionally, the KOBAS software 3.0 [81] was utilized to assess the statistical enrichment of these DEGs in KEGG pathways (http://www.genome.jp/kegg/, accessed on 15 December 2023). After reconstructing the transcript using Stringtie, some genes that were identified in the current sequencing results but not included in the reference genome (or reference gene collection) were detected. These genes were then annotated by comparing them with sequences in the KEGG and GO databases. These genes are defined as novel genes.

#### 4.2.4. Quantitative Real-Time PCR (qRT-PCR) Validation

To verify the accuracy of the RNA-Seq results, 10 DEGs were selected for qRT-PCR analysis. Total RNA (1 μg) from ZQ and CK samples was reverse transcribed using the PrimeScript RT reagent Kit with gDNA Eraser (Perfect Real Time) (Takara, Otsu, Japan) according to the manufacturer’s instructions. The qRT-PCR conditions were as follows: an initial denaturation at 95 °C for 30 s, followed by 40 cycles of 95 °C for 5 s and 60 °C for 5 s. The reactions were performed using the ABI PRISM 7500 Fast Real-Time PCR System (Applied Biosystems, Foster City, CA, USA). Data analysis was conducted using the 7500 system sequence detection software (version 1.4) [82]. Beta-tubulin was used as a reference gene [83]. The primers used for qRT-PCR validation of the DEGs are listed in Appendix A.

### 4.3. Metabolome Analysis

#### 4.3.1. Metabolite Extraction

The plant tissues (80 mg of leaves) were quickly frozen in liquid nitrogen and then ground into fine powder using a mortar and pestle. For metabolite extraction, 1000 μL of a methanol/acetonitrile/H_2_O mixture (2:2:1, *v*/*v*/*v*) were added to the homogenized solution. The mixture was centrifuged for 15 min at 14,000 g and 4 °C. The supernatant was then dried using a vacuum centrifuge. For LC-MS analysis, the samples were re-dissolved in 100 μL of an acetonitrile/water solvent (1:1, *v*/*v*). To monitor the stability and repeatability of instrument analysis, quality control (QC) samples were prepared by pooling 10 μL of each sample and analyzed together with the other samples. The raw sequencing reads containing adapters or low-quality bases were filtered using fastp software (version 0.18.0). The filtered reads were then aligned to the reference genome (version: GCF_002738345.1) using HISAT2 software 2.2.1. Following alignment, gene expression quantification was performed using RSEM 1.3.3.

#### 4.3.2. LC-MS/MS Analysis

The analysis was conducted using an ultra-high-performance liquid chromatography (UHPLC) system, specifically the 1290 Infinity LC from Agilent Technologies, coupled with a quadrupole time-of-flight mass spectrometer (AB Sciex TripleTOF 6600) at Shanghai Applied Protein Technology Co., Ltd, Shanghai, China.

For HILIC separation, samples were analyzed using a 2.1 mm × 100 mm ACQUITY UPLC BEH 1.7 µm column (Waters Corporation, Milford, MA, USA). In both ESI positive and negative modes, the mobile phase consisted of A = 25 mM ammonium acetate and 25 mM ammonium hydroxide in water, and B = acetonitrile. The gradient started at 85% B for 1 min, linearly decreased to 65% in 11 min, then reduced to 40% in 0.1 min and maintained for 4 min, before increasing back to 85% in 0.1 min, followed by a 5 min re-equilibration period.

For RPLC separation, a 2.1 mm × 100 mm ACQUITY UPLC HSS T3 1.8 µm column (Waters, Ireland) was utilized. In ESI positive mode, the mobile phase consisted of A = water with 0.1% formic acid and B = acetonitrile with 0.1% formic acid. In ESI negative mode, the mobile phase was composed of A = 0.5 mM ammonium fluoride in water and B = acetonitrile. The gradient started at 1% B for 1.5 min, then linearly increased to 99% in 11.5 min, and was maintained for 3.5 min. It was then reduced to 1% in 0.1 min, followed by a 3.4 min re-equilibration period. The gradients were run at a flow rate of 0.3 mL/min, and the column temperature was kept constant at 25 °C. A 2 µL aliquot of each sample was injected.

The ESI source conditions were set as follows: Ion Source Gas 1 (Gas1) at 60, Ion Source Gas 2 (Gas2) at 60, curtain gas (CUR) at 30, source temperature at 600 °C, and IonSpray Voltage Floating (ISVF) at ±5500 V. In MS-only acquisition, the instrument was configured to acquire data over the m/z range of 60–1000 Da, with an accumulation time for TOF MS scan set at 0.20 s/spectra. In auto MS/MS acquisition, the instrument was set to acquire data over the m/z range of 25–1000 Da, with an accumulation time for product ion scan set at 0.05 s/spectra. The product scan was acquired using information-dependent acquisition (IDA) with the high-sensitivity mode selected. The parameters were set as follows: the collision energy (CE) was fixed at 35 V with a range of ±15 eV; declustering potential (DP) at 60 V (+) and −60 V (−); isotopes within 4 Da were excluded, and the number of candidate ions to monitor per cycle was set to 10.

#### 4.3.3. Metabolite Detection

Based on Mass Bank, Metlin, MoNA, HMDB, and a self-built library, Analyst 1.6.3 and secondary spectral information were utilized to process the mass spectrum data. The metabolites were qualitatively analyzed, and their quantitative analysis was performed using triple quadrupole mass spectrometry in multi-reaction monitoring mode. The raw MS data were converted to MzXML files using ProteoWizard MSConvert before being imported into freely available XCMS software 3.18.0. For peak picking, the following parameters were used: centWave m/z = 10 ppm, peakwidth = c (10, 60), prefilter = c (10, 100). For peak grouping, bw = 5, mzwid = 0.025, minfrac = 0.5 were used. CAMERA (Collection of Algorithms of MEtabolite pRofile Annotation) was sued for annotation of isotopes and adducts. In the extracted ion features, only the variables having more than 50% of the nonzero measurement values in at least one group were kept. Compound identification of metabolites was performed by comparing accuracy m/z value (<10 ppm) and MS/MS spectra with an in-house database established with available authentic standards. The missing data were filled by KNN (K-Nearest Neighbor) method, and the extreme values were deleted. Finally, the total peak area of the data was normalized to ensure the parallelism between samples and metabolites.

The PCA analysis was performed using the R language package gmodels (version 2.18.1). PCA plots were employed to analyze the variability both between and within the visualized groups. Orthogonal Partial Least Squares Discriminant Analysis (OPLS-DA) was utilized to assess the reliability of the application model. Based on the OPLS-DA results, differential accumulation metabolites were screened using thresholds of VIP ≥ 1 and Student’s *t*-test *p* < 0.05, combining Variable Importance Projection (VIP) and Fold Change (FC) values from the OPLS-DA model. The GO and KEGG databases were used for the functional annotation of these DAMs.

### 4.4. Metabolome and Transcriptome Association Analysis

#### 4.4.1. Pathway Model

KEGG pathway maps link genomic or transcriptomic data of genes to the chemical structures of endogenous molecules, thereby providing a method for integrating the analysis of genes and metabolites. In this study, all DEGs and metabolites were mapped to the KEGG pathway database to identify their connections within metabolic pathways.

#### 4.4.2. O2PLS Model

To integrate the transcriptomic and metabolomic data, we conducted a Two-way Orthogonal PLS (O2PLS) analysis [84]. This method decomposes the variation present in the two data matrices into three components: the joint variation between the two datasets, the orthogonal variation unique to each dataset, and noise. The model assumes that latent variables are responsible for the variation in both the joint and orthogonal parts. O2PLS models were calculated using the OmicsPLS package in R 1.0.2 [85]. To determine the optimal number of components, an alternative cross-validation procedure was employed [86]. The best models were then used for integration analysis.

#### 4.4.3. Pearson Model

Pearson correlation coefficients were computed to integrate metabolome and transcriptome data. Gene–metabolite pairs were ranked based on the descending order of their absolute correlation coefficients. The top 30 gene–metabolite pairs were selected for heatmap analysis using the pheatmap package 1.0.12 in the R programming environment, with the package version available at http://CRAN.R-project.org/package=pheatmap, accessed on 9 January 2024. Furthermore, the top 250 gene–metabolite pairs, exhibiting an absolute Pearson correlation greater than 0.5, were utilized for constructing a metabolite–transcript network analysis using the igraph package in R 1.3.5 [86,87].

## Figures and Tables

**Figure 1 ijms-26-02586-f001:**
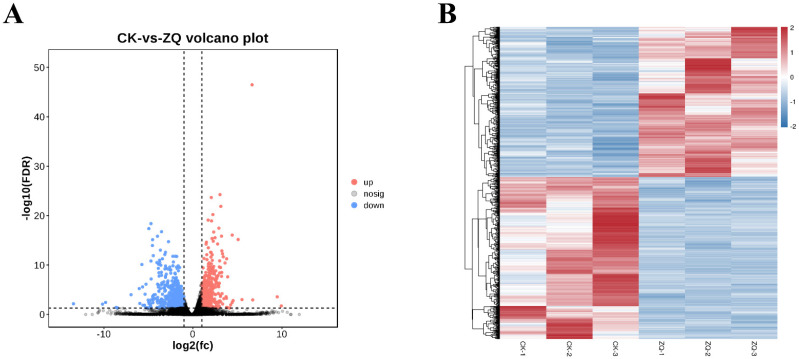
**Differential expression analysis.** (**A**) **Volcano plot of DEGs.** Each point in the volcano plot represents a gene. Blue points indicate significantly downregulated genes, red points indicate significantly upregulated genes, and black points represent genes with no significant difference in expression among DEGs, providing a clear visualization of gene expression significance changes. (**B**) **Clustergram of DEGs expression patterns.** Displays the expression levels of DEGs across six samples (ZQ1-ZQ3 and CK1-CK3). Different lines represent different samples, different rows represent different genes, and color intensity indicates the log value of gene expression quantity (FPKM) with a base of 2, vividly presenting the similarities and differences in gene expression.

**Figure 2 ijms-26-02586-f002:**
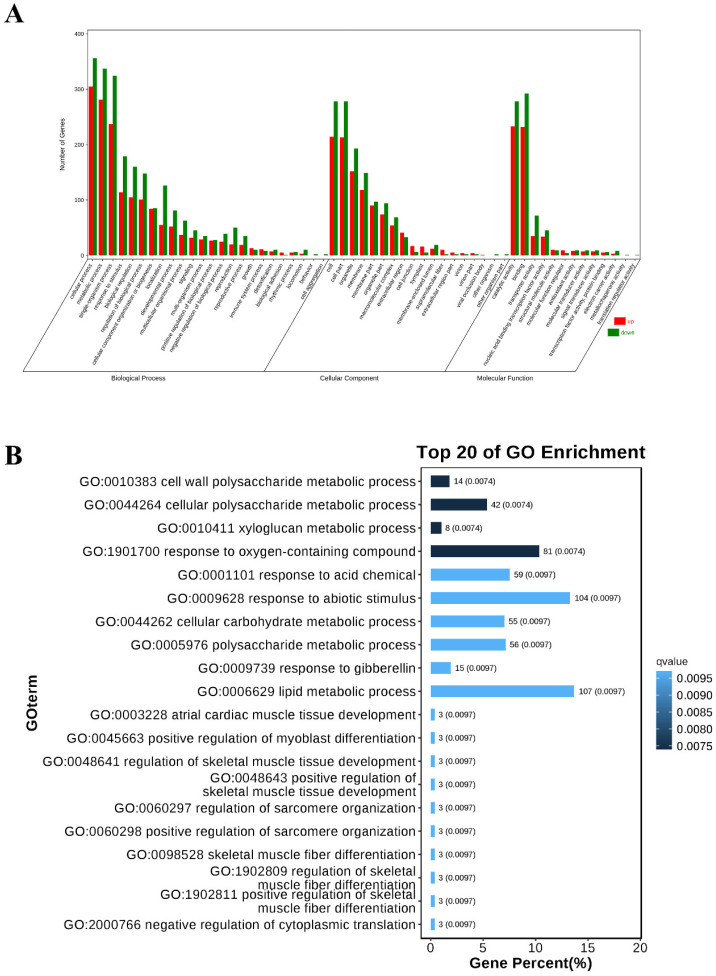
**GO enrichment analysis of DEGs.** (**A**) **Statistical diagram of GO secondary node annotation of DEGs.** The X-axis represents GO categories, and the Y-axis represents the number of genes, clearly showing the distribution of DEGs across various biological processes, cellular components, and molecular functions. (**B**) **Top 20 GO enrichment analysis.** The top 20 GO terms with the smallest Q values are visualized. The Y-axis is the GO term, and the X-axis is the percentage of these GO terms in the total number of differential genes. The darker the color, the smaller the Q value, and the values on the columns clearly indicate the number and Q value of each GO term, highlighting the gene function enrichment areas with significant biological meaning in the study.

**Figure 3 ijms-26-02586-f003:**
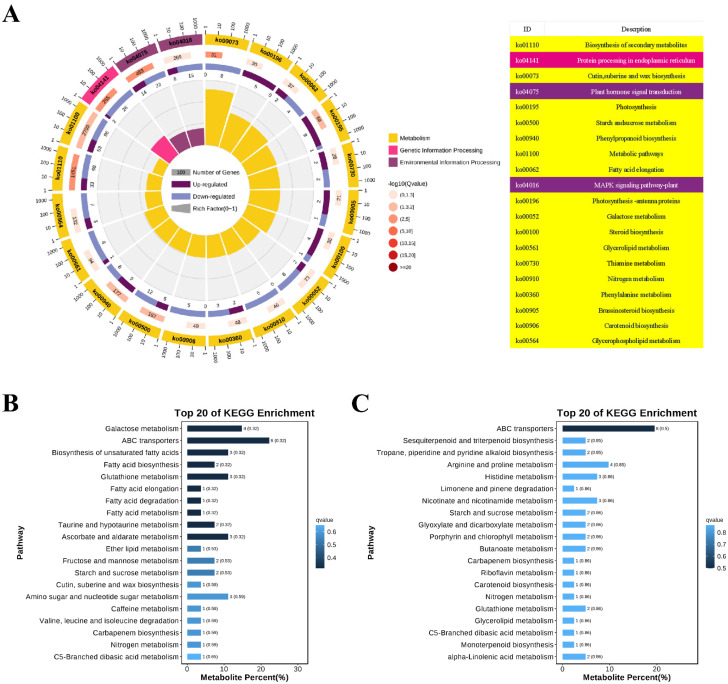
**KEGG enrichment analysis.** (**A**) **Top 20 KEGG enrichment of DEGs.** The chart consists of multiple concentric circles. The outermost circle displays the top 20 KEGG-enriched pathways, and the outer circle indicates the coordinate of the number of differential genes. The second circle shows the number and Q value of this pathway in the differential gene background. The more differential genes there are, the longer the bar shape, and the smaller the Q value, the redder the color, intuitively presenting the significance of the pathway. The third circle is a bar chart of up-down-differential gene ratio, with dark purple representing up–down-differential gene ratio and light purple representing down–down-differential gene ratio, revealing the specific direction of gene expression changes in the pathway. The innermost circle shows the Rich Factor value of each pathway (the number of differential genes in that pathway divided by the total number of genes in that pathway), with each background grid line cell representing 0.1, providing precise quantification for assessing the enrichment degree of the pathway. (**B**) **Top 20 KEGG enrichment of DAMs (NEG).** The KEGG enrichment analysis results of metabolites in negative ion mode, showing the enrichment of metabolites in different biological pathways. (**C**) **Top 20 KEGG enrichment of DAMs (POS).** The KEGG enrichment analysis results of metabolites in positive ion mode, further supplementing the enrichment information of metabolites in biological pathways.

**Figure 4 ijms-26-02586-f004:**
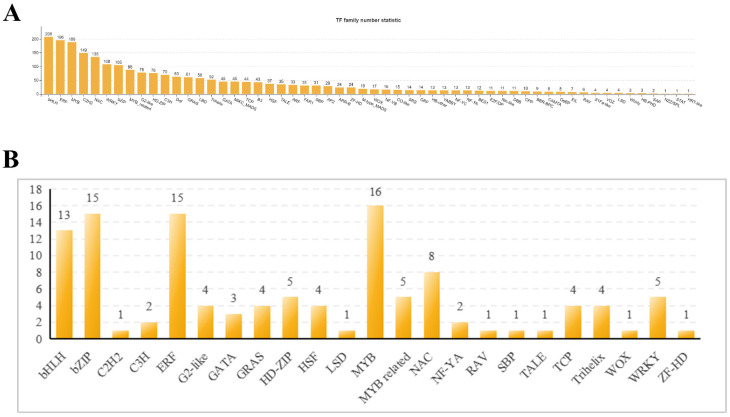
**Analysis of transcription factors in pumpkin during Sc infection.** (**A**) **Families of total transcription factors.** Comprehensively displays the classification of all transcription factors in pumpkin during the infection period, providing a foundation for understanding the regulatory role of transcription factors in the pathogen infection process. (**B**) **Families of significantly enriched transcription factors.** Focuses on significantly enriched transcription factor families, highlighting those that may play a key regulatory role in pathogen infection, pointing the way for further research into their functional mechanisms.

**Figure 5 ijms-26-02586-f005:**
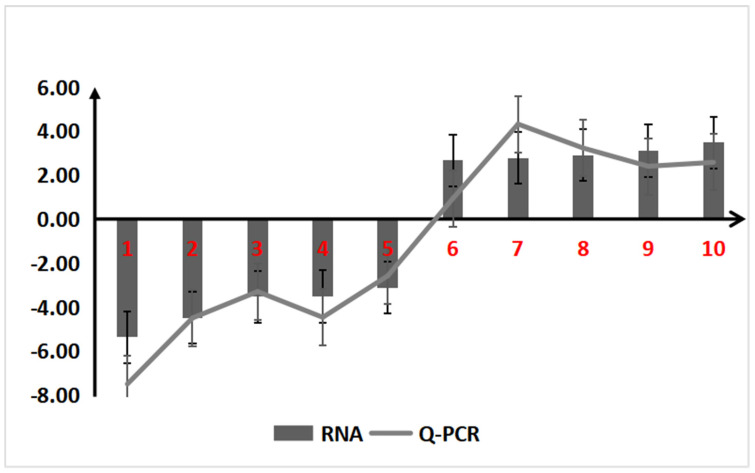
Comparison of relative expression levels of 10 DGEs between CK and ZQ determined by qRT-PCR versus RPKM values from RNA-seq. The qRT-PCR ratio (determined by the log2−ΔΔCT values from ZQ divided by that from CK) is compared with the RPKM ratio (determined by the log2 of the RPKM value from ZQ divided by that of CK), verifying the reliability and accuracy of RNA-seq data and ensuring the scientific nature of subsequent analysis results.

**Figure 6 ijms-26-02586-f006:**
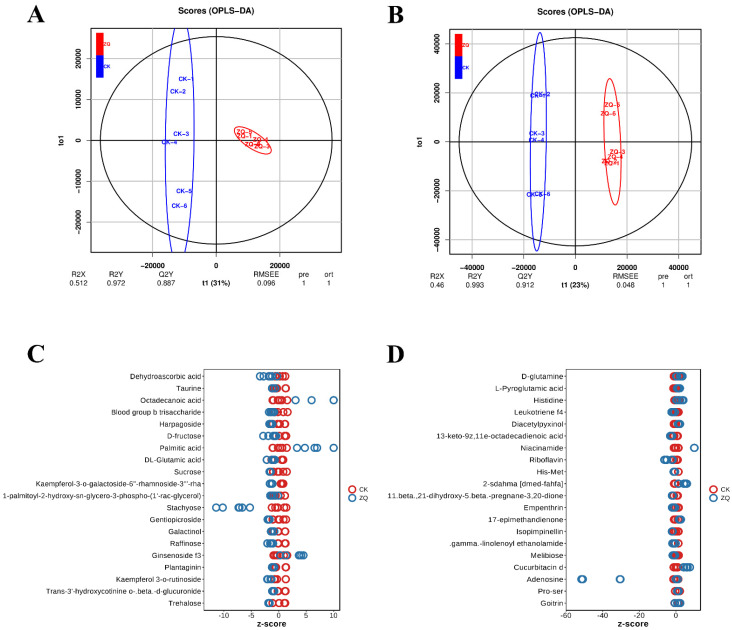
**OPLS-DA and Z-score results of DAMs analysis.** (**A**) **OPLS-DA scatter plot of resistant line in NEG ion mode.** The OPLS-DA model is used to analyze metabolite data in negative ion mode, clearly showing the distribution characteristics and group differences of resistant line samples in the metabolite space. (**B**) **OPLS-DA scatter plot of resistant line in POS ion mode.** The OPLS-DA analysis of metabolite data in positive ion mode further supplements the distribution information of resistant line samples, providing a basis for comprehensively understanding their metabolic characteristics. (**C**) **Z-score diagram of CK vs. ZQ group in NEG ion mode.** The horizontal axis represents the standardized value of metabolite abundance after z-score, and the vertical axis is each behavior’s metabolite. Red represents CK (control sample), blue represents ZQ (treatment group sample), and each circle represents a sample. Z-score < 0 indicates downregulated metabolites, and Z-score > 0 intuitively shows the upregulation and downregulation of metabolites between CK and ZQ groups in negative ion mode. (**D**) **Z-score diagram of CK vs. ZQ group in POS ion mode.** Similarly, the horizontal axis is z-score, the vertical axis is metabolites, red represents CK, blue represents ZQ, and each circle represents a sample. Z-score < 0 indicates downregulated metabolites, and Z-score > 0 shows the upregulation and downregulation of metabolites between CK and ZQ groups in positive ion mode, providing an intuitive visualization for comparing metabolic changes between the two groups.

**Figure 7 ijms-26-02586-f007:**
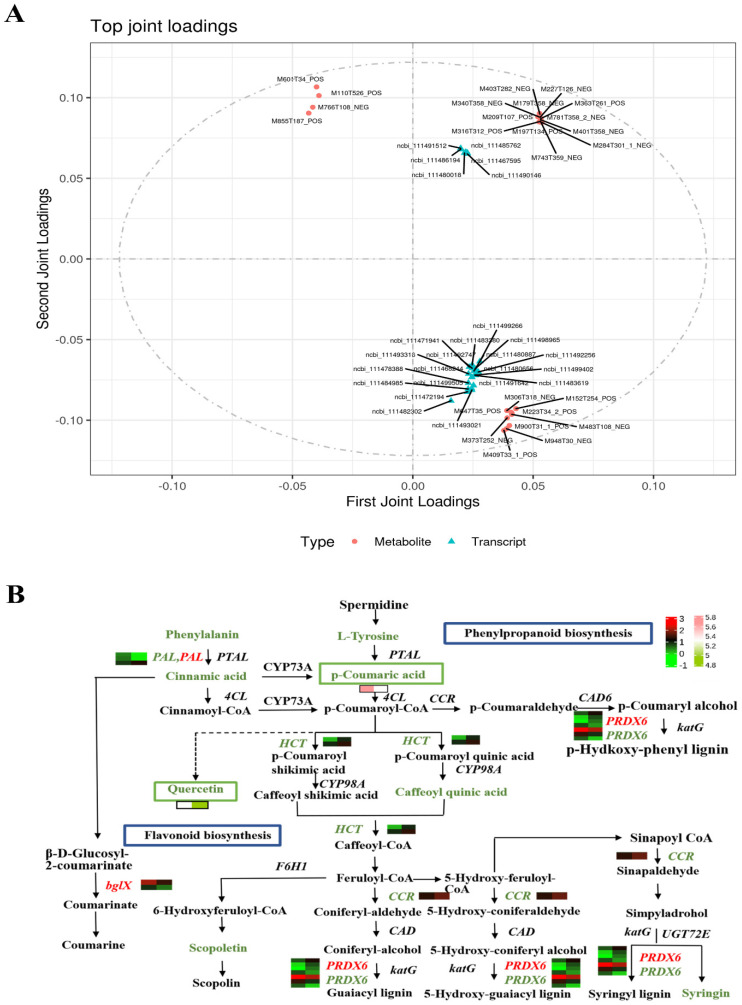
**Analysis of correlation between DEGs and DAMs.** (**A**) **Metabolite and gene correlation payload diagram.** The payload value indicates the explanatory ability of the variable (metabolite/gene) in each component (i.e., the contribution to the differences between groups), with positive and negative payload values indicating positive or negative correlation with another group of studies. The greater the absolute value of the load, the stronger the correlation, and the diagram intuitively reveals the strength and direction of the correlation between metabolites and genes, providing a basis for understanding their synergistic roles in biological processes. (**B**) **Phenylpropanoid biosynthesis and flavonoid biosynthesis pathways.** Detailed display of the changes in phenylpropanoid biosynthesis and flavonoid biosynthesis pathways in pumpkin during pathogen invasion, highlighting the key role of these pathways in plant defense responses.

**Table 1 ijms-26-02586-t001:** CK-vs-ZQ transcriptome and proteome association number statistics.

Type	mRNAs	Proteins
All Genes	26,596	8146
Diff Genes	1176	370
Up Genes	566	70
Down Genes	610	300

Note: All genes refer to genes/proteins that can be detected in this project, and Diff Genes refer to genes/proteins with significant differences. The threshold of significantly different genes was reduced by two times to the default threshold, and FDR(or *p*-value) was less than 0.05. The threshold of significantly different proteins was reduced by 1.5 times to the default threshold, and P(or Q) was less than 0.05.

## Data Availability

Data is contained within the article and Appendix A.

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
