# Peer review of "Integrated Transcriptome and Metabolome Analysis Elucidates the Defense Mechanisms of Pumpkin Against Gummy Stem Blight"

_ijms, 2025, doi:10.3390/ijms26062586_

Round 1
Reviewer 1 Report
Comments and Suggestions for Authors
- Line 129: Distilled water was used as a control, but it is unclear whether the water was sterile. This should be explicitly stated.
- RNA Extraction: There is no description of the RNA extraction method, which is a critical omission.
- Reads QC and Assembly: The methodology for read quality control (QC) and assembly is missing. These details appear in the results section, which seems misplaced and should instead be included in the methods section.
- Line 217: The information provided on metabolite detection is insufficient for validating the findings. The authors mention the tools used but do not specify their versions or clarify the role of each tool in the analysis pipeline.
- Line 343: The authors claim to have discovered 858 new genes. However, since they performed reference-based transcriptome assembly, it is unclear how novel genes could be identified. This method can only map transcripts to existing annotations and cannot discover genes beyond those already present in the reference genome.
- Page 447: The presentation of POS and NEG metabolites needs improvement. As it stands, they appear as two distinct sample sets, despite being derived from the same samples and differing only in ionization mode. To enhance clarity and provide a more comprehensive interpretation, both datasets should be combined and analyzed together.
- Transcriptome Data: The transcriptome data should be deposited in a public repository, and the corresponding accession numbers must be provided.
- Methods and Results: The authors have mixed methodological details with results, which creates confusion. These sections should be clearly separated for better readability and logical flow.
Author Response
Comments 1: [Line 129: Distilled water was used as a control, but it is unclear whether the water was sterile. This should be explicitly stated.] |
Response 1: [Thank you for pointing this out. We have added the necessary details to line 132 of the manuscript.] |
Comments 2: [RNA Extraction: There is no description of the RNA extraction method, which is a critical omission.] |
Response 2: [Thank you for pointing this out. We have revised the manuscript to include additional details about the RNA extraction method in line 142.] |
Comments 3: [Reads QC and Assembly: The methodology for read quality control (QC) and assembly is missing. These details appear in the results section, which seems misplaced and should instead be included in the methods section.] |
Response 3: [Agree. Therefore, We have supplemented the missing sections related to Reads QC and Assembly methods, and revised the method-related content in the results section. These revisions are marked at lines 190 and 444 of the manuscript. ] |
Comments 4: [Line 217: The information provided on metabolite detection is insufficient for validating the findings. The authors mention the tools used but do not specify their versions or clarify the role of each tool in the analysis pipeline.] |
Response 4: Agree. We have supplemented the information provided regarding metabolite detection in the Methods section, while also specifying the corresponding versions and usage in line 234. |
Comments 5: [Line 343: The authors claim to have discovered 858 new genes. However, since they performed reference-based transcriptome assembly, it is unclear how novel genes could be identified. This method can only map transcripts to existing annotations and cannot discover genes beyond those already present in the reference genome.] |
Response 5: [Thank you for pointing this out. We have added the relevant information in line 171 of the manuscript. In this article, the "novel genes" refer to genes that start with "MSTRG." These genes are not native to the genome but were identified based on sequencing data. Specifically, some sequencing fragments were found to align with intergenic regions in the genome, suggesting they might be transcriptions. Using the StringTie software, these transcripts were reconstructed, and their coding potential was assessed. Those with coding potential were defined as novel genes. These findings are based on omics data predictions and require experimental verification. Essentially, this work represents a supplement to the existing gene annotations in the genome.] |
Comments 6: [Page 447: The presentation of POS and NEG metabolites needs improvement. As it stands, they appear as two distinct sample sets, despite being derived from the same samples and differing only in ionization mode. To enhance clarity and provide a more comprehensive interpretation, both datasets should be combined and analyzed together.] |
Response 6: [Thank you for pointing this out. Combining datasets and performing differential analysis again may alter the set of differential metabolites. However, if the metabolic datasets are merged first before conducting OPLS-DA analysis, it will change the variable importance on projection (VIP) scores of the metabolites. As VIP scores are the criterion for selecting differential metabolites, this alteration could modify the set of differential metabolites, which may lead to substantial revisions in both results and conclusions.Thank you once again for your suggestion. We will place greater emphasis on this point in our future research.] |
Comments 7: [Transcriptome Data: The transcriptome data should be deposited in a public repository, and the corresponding accession numbers must be provided.] |
Response 7: [Thank you for pointing this out. We have submitted the data to Genome Sequence Archive(GSA)database and added the corresponding content to the manuscript. ] |
Comments 8: [The authors have mixed methodological details with results, which creates confusion. These sections should be clearly separated for better readability and logical flow.] |
Response 8: [Agree. We have made corresponding revisions to the manuscript according to the reviewers' suggestions.] |

Reviewer 2 Report
Comments and Suggestions for Authors
The manuscript titled "Integrated Transcriptome and Metabolome Analysis Elucidates the Defense Mechanisms of Pumpkin Against Gummy Stem Blight" presents a multi-omics approach to understanding the molecular defense mechanisms in pumpkin against Gummy Stem Blight (GSB) caused by Stagonosporopsis cucurbitacearum. The study employs transcriptomics and metabolomics analyses to uncover differentially expressed genes (DEGs) and differentially accumulated metabolites (DAMs), revealing pathways involved in plant-pathogen interactions, phytohormone signaling, and phenylpropanoid biosynthesis. The findings provide valuable insights into resistance mechanisms that could contribute to future breeding strategies for GSB-resistant pumpkin varieties.
1. The integration of transcriptomics and metabolomics enhances the study’s depth, allowing a more comprehensive understanding of the pumpkin's defense response. The use of O2PLS and KEGG enrichment analyses strengthens the study's scientific rigor.
2. The authors provide a thorough description of plant material selection, artificial inoculation, RNA sequencing, metabolomics analysis, and statistical validation methods. The inclusion of qRT-PCR validation adds credibility to the transcriptomic findings.
3. The study successfully identifies key metabolic pathways, including phenylpropanoid biosynthesis and flavonoid biosynthesis, that play a role in pathogen resistance. The identification of p-coumaric acid and quercetin as potential defense metabolites is noteworthy.
4. The application of advanced bioinformatics tools, such as differential expression analysis, correlation networks, and pathway enrichment, ensures that the study's conclusions are well-supported. The consistency between transcriptomic and metabolomic data enhances the study's reliability.
5. The findings contribute valuable knowledge for breeding disease-resistant pumpkin varieties, with potential applications in sustainable agriculture and disease management.
6. While the study identifies crucial genes and metabolites involved in the resistance response, it lacks functional validation (e.g., gene knockdown or overexpression studies) to confirm their precise roles in GSB resistance.
7. The study focuses solely on pumpkin, but a comparison with other cucurbit species could provide a broader context for understanding conserved and species-specific defense mechanisms.
8. The authors rely on OPLS-DA for metabolite selection, but additional statistical tests such as ANOVA or Wilcoxon rank-sum tests could strengthen the reliability of the identified DAMs.
9. While the discussion effectively interprets the results, it could be improved by critically assessing potential study limitations, alternative explanations, or inconsistencies in the data.
10. The study primarily focuses on mechanistic insights, but a discussion on how these findings can be translated into real-world agricultural practices (e.g., breeding programs, biocontrol strategies) would enhance the study's impact.
The manuscript presents significant findings on pumpkin resistance to GSB using an integrated omics approach. The study is well-executed, and the data analysis is thorough, making it a valuable contribution to plant-pathogen interaction research. However, additional experimental validation of key findings and a more critical discussion would further strengthen the manuscript.
Decision: Minor Revisions Required
I recommend the authors address the concerns regarding functional validation, statistical robustness, and practical applications before acceptance.
Author Response
Comments 1: [The integration of transcriptomics and metabolomics enhances the study’s depth, allowing a more comprehensive understanding of the pumpkin's defense response. The use of O2PLS and KEGG enrichment analyses strengthens the study's scientific rigor.] |
Response 1: [Thank you very much for your valuable advice.]
|
Comments 2: [The authors provide a thorough description of plant material selection, artificial inoculation, RNA sequencing, metabolomics analysis, and statistical validation methods. The inclusion of qRT-PCR validation adds credibility to the transcriptomic findings.] |
Response 2: [Thank you very much for your valuable advice.]
|
Comments 3: [The study successfully identifies key metabolic pathways, including phenylpropanoid biosynthesis and flavonoid biosynthesis, that play a role in pathogen resistance. The identification of p-coumaric acid and quercetin as potential defense metabolites is noteworthy.] |
Response 3: [Thank you very much for your valuable advice.]
|
Comments 4: [The application of advanced bioinformatics tools, such as differential expression analysis, correlation networks, and pathway enrichment, ensures that the study's conclusions are well-supported. The consistency between transcriptomic and metabolomic data enhances the study's reliability.] |
Response 4: Thank you very much for your valuable advice.
|
Comments 5: [The findings contribute valuable knowledge for breeding disease-resistant pumpkin varieties, with potential applications in sustainable agriculture and disease management.] |
Response 5: [Thank you very much for your valuable advice.]
|
Comments 6: [While the study identifies crucial genes and metabolites involved in the resistance response, it lacks functional validation (e.g., gene knockdown or overexpression studies) to confirm their precise roles in GSB resistance.]
|
Response 6: [Thank you for bringing this to our attention. We have included a section on validation of gene function in the discussion portion of our research manuscript and will incorporate additional related content in future research.]
|
Comments 7: [The study focuses solely on pumpkin, but a comparison with other cucurbit species could provide a broader context for understanding conserved and species-specific defense mechanisms.] |
Response 7: [Thank you for pointing this out. We added some content on comparison with other cucurbit species regarding species-specific defense mechanisms in the discussion part of our research manuscript.]
|
Comments 8: [The authors rely on OPLS-DA for metabolite selection, but additional statistical tests such as ANOVA or Wilcoxon rank-sum tests could strengthen the reliability of the identified DAMs.]
|
Response 8: [Agree. In addition to the criteria for screening differential metabolites, we also employed other statistical methods, such as: in the OPLS-DA model, VIP ≥ 1 and Student’s t-test P < 0.05, and revisions were made in the manuscript.]
|
Comments 9: [While the discussion effectively interprets the results, it could be improved by critically assessing potential study limitations, alternative explanations, or inconsistencies in the data.]
|
Response 9: [Agree. We have made corresponding revisions in the manuscript according to the reviewers' suggestions.]
|
Comments 10: [The study primarily focuses on mechanistic insights, but a discussion on how these findings can be translated into real-world agricultural practices (e.g., breeding programs, biocontrol strategies) would enhance the study's impact.]
|
Response 10: [Agree. We have made corresponding revisions in the manuscript according to the reviewers' suggestions.] |
